# High temporal resolution records of the velocity of Hansbreen, a tidewater glacier in Svalbard

Małgorzata Błaszczyk[1], Bartłomiej Luks[2], Michał Pętlicki[3], Dariusz Puczko[4], Dariusz Ignatiuk[1], Michał Laska[1], Jacek Jania[1], Piotr Głowacki[2]

[1]Faculty of Natural Sciences, University of Silesia in Katowice, Katowice, 40-007, Poland
[2]Institute of Geophysics, Polish Academy of Sciences, Warszawa, 01-452, Poland
[3]Faculty of Geography and Geology, Jagiellonian University, Kraków, 30-387, Poland
[4]Institute of Biochemistry and Biophysics, Polish Academy of Sciences, Warszawa, 02-106, Poland

*Correspondence to*: Małgorzata Błaszczyk (malgorzata.blaszczyk@us.edu.pl)

**Abstract.** Monitoring changes in glacial dynamics is essential for understanding the environmental response to accelerated climate warming in the Arctic. However, geodetic surveys in polar regions continue to present considerable challenges because of the harsh environmental conditions and the polar night. This study records a 14–year–long time series (2006–2019) of GNSS surveys of the positions of 16 ablation stakes distributed across Hansbreen, a tidewater glacier in southern Svalbard. The measurements were conducted with an exceptionally high temporal resolution, from about one week to about one month, and covering altitudes ranging from 0 to 490 m asl. The position of one stake was surveyed every day. The primary data products consist of the stake coordinates and velocities. Time series of annual and seasonal velocities are also provided. This dataset may be a subject of further studies of glacier dynamics in relation to the long–term and seasonal impact of climate change on ice flow in the region. It also offers unique material for tuning numerical models of glacier dynamics and for validating satellite–derived products such as velocity and digital elevation models. The dataset described here has been made publicly available through the Zenodo repository: https://doi.org/10.5281/zenodo.8289380 (Błaszczyk et al., 2023).

## 1. Introduction

The study of ice mass dynamics in light of climate changes is a major research topic (Benn and Evans, 1998; Lemke et al., 2007; Strozzi et al., 2017; Meredith et al., 2019; Van Wychen et al., 2016). For more than two decades, in situ measurements of glacier surface velocities with GNSS (global navigation satellite system, Vieli et al., 2000, 2004; Stober and Hepperle, 2007; Machío et al., 2017) have been among the most valuable sources of information for the study of glacier dynamics. The surface velocity is the sum of two components – ice deformation and basal slip (Nye, 1952; Weertman, 1957). The forces governing flow depend on several factors, such as temperature, the debris content of the ice, bed roughness and water pressure (Benn and Evans, 1998). As there are still few data on flow rates at depth, the surface velocity, the easiest component of flow to measure (Cuffey and Patterson, 2010), is used for constructing numerical models of glaciers (Vieli et al., 2004; Rückamp et al., 2010; Otero et al., 2017; Vallot et al., 2017; Koziol and Arnold, 2018; Klein et al., 2020; Rosier and Gudmundsson, 2020).

Few sets of long–term in situ measurements of glacier velocities in polar regions are available, as geodetic measurements are time–consuming, logistically difficult to perform, and expensive (Stober and Hepperle, 2019). Nonetheless, they are especially important for acquiring knowledge and an understanding of the mechanism of mass circulation in the glacier system. They are

also useful for determining the volume of ice lost by glaciers flowing into the sea as a result of frontal ablation, a process that is not often taken into account in mass balance studies of such glaciers. The Svalbard glaciers where long–term velocity measurements have been studied are Kongsvegen (Voigt, 1965; Vivet and Lliboutry, 1978; Lefauconnier, 1987; Lefauconnier et al., 1994; Hagen et al., 2005), Kronebreen (Schellenberger et al., 2015; Vallot et al., 2017), and Nordenskiöldbreen (den Ouden et al., 2010; van Pelt et al., 2018). Other in situ velocity measurements of tidewater glaciers in Svalbard have usually

been limited to just a few years, e.g. five years on Austfonna (Dunse et al., 2015), and two on Holtedahlfonna (Bahr, 2015).

This study aims to present a data set of in situ glacier surface velocities of Hansbreen (Svalbard), collected for the 14–year period 2006–2019, obtained from the mass balance stakes network surveyed using GNSS. This continuous measurement record presents velocity of Arctic glacier with unique and exceptionally high temporal resolution, around one week for five stakes

and around one month for eleven stakes. To our knowledge, Hansbreen was the first Svalbard tidewater glacier for which ice flow velocity was measured with dual-frequency GPS on a regular basis resulting in exceptionally high measurement accuracy. The presented data also fill the gap in discontinuous satellite-derived glacier flow velocities before beginning of Sentinel-1 mission in 2014. Hansbreen is one of the few well–studied tidewater glaciers in Svalbard. The Polish Polar Station Hornsund, located in the vicinity of the glacier, provides the logistics for permanent measurements. Hansbreen is also one of the key

glaciers in the World Glacier Monitoring Service (WGMS, 2021; https://wgms.ch, latest access: 30 June 2023), and its annual mass balance has been measured since 1989. Additionally, the surface velocity near the terminus of Hansbreen was measured by terrestrial photogrammetry from 1982 to 1991 (Jania and Kolondra, 1982; Vieli et al., 2000) and GNSS surveying (Vieli et al., 2004). Therefore, as a well-studied, polythermal, non-surging tidewater glacier of medium size, Hansbreen can be representative for the entire Svalbard Archipelago. Presented velocity data set can serve as input data for numerical models of

glacier dynamics and processes at glacier termini (see, e.g. Otero et al., 2017; de Andrés et al., 2018). In situ measurements are invaluable for validating satellite–derived velocities obtained with methods like differential interferometric synthetic aperture radar and SAR offset tracking (Schellenberger et al., 2015; Błaszczyk et al., 2019a; Solgaard et al., 2021). The height of the glacier can be used for validating numerous digital elevation models derived from different remote sensing techniques (Jawak and Luis, 2012; Berthier et al., 2014; Schröder et al., 2017; Błaszczyk et al., 2019b). As climate warming in Hornsund

took place more than six times as fast as the global average in the period 1979–2018 (Wawrzyniak and Osuch, 2020), such long–term observations permit a better insight into glacier dynamics in a rapidly changing environment. Our understanding of the physics of glacier motion is still incomplete (Benn and Evans, 1998), especially regarding connection between ice flow and surface runoff. This study provides dataset that enable a detailed assessment of spatial and temporal changes in flow of Svalbard tidewater glacier in the context of surface runoff, both in short-term and interannual time scale.

## 2. Study area

Hansbreen (15°37'E, 77°05'N) is a polythermal tidewater glacier terminating in the Hornsund fjord, southern Svalbard (Fig. 1). In 2015, the glacier covered an area of 51.3 km$^2$ and had an active calving front 1.7 km wide (Błaszczyk et al., 2019a). The Hansbreen system was described in detail by Grabiec et al. (2012). It consists of a main glacier trunk and four major western tributary glaciers: Staszelisen, Deileggbreen, Tuvbreen and Fuglebreen. The surface slope is 1.8° along the centreline of Hansbreen and up to 5.6° along the centrelines of the tributary glaciers. The ice divide between Hansbreen and Vrangpeisbreen is well defined and located at around 490 m a.s.l, the highest part of the main trunk. However, the boundary between Hansbreen and its eastern neighbour Paierlbreen is more difficult to define owing to the transfluence of ice from the accumulation field to Kvitungisen, a tributary of Paierlbreen. The terminus retreat has been studied by Błaszczyk et al. (2013, 2021). The terminus of Hansbreen has been retreating since 1899, although there have been sporadic episodes of advance. From 1991 to 2015, the front retreated 917 m.

Hansbreen is the westernmost glacier in Hornsund, the southernmost fjord of Svalbard Archipelago. The hydrographic conditions along the west coast of Spitsbergen are defined by the cold water of Spitsbergen Current and warmer water of West Spitsbergen Current (Promińska et al., 2018). According to Nilsen et al. (2016), in recent years warm water from the West Spitsbergen Current flows more often into the western fjords, even in winter. Further, the glaciers of southern and central Spitsbergen have the most negative climate balance on Svalbard (Schmidt et al., 2023; van Pelt et al., 2019), and models predict the greatest glacier mass loss in the southern part of the archipelago also in the 2019-2060 period (van Pelt et al., 2021). Furthermore, global mass balance models (excluding the Greenland and Antarctic ice sheets) point to Svalbard as the region with the greatest negative mass balance by the end of the 21st century (Huss and Hock, 2015). In light of these studies, understanding a response of Hansbreen dynamics to climate warming is important for predicting glacial dynamic processes in the rest of the archipelago, as well as for other Arctic areas.

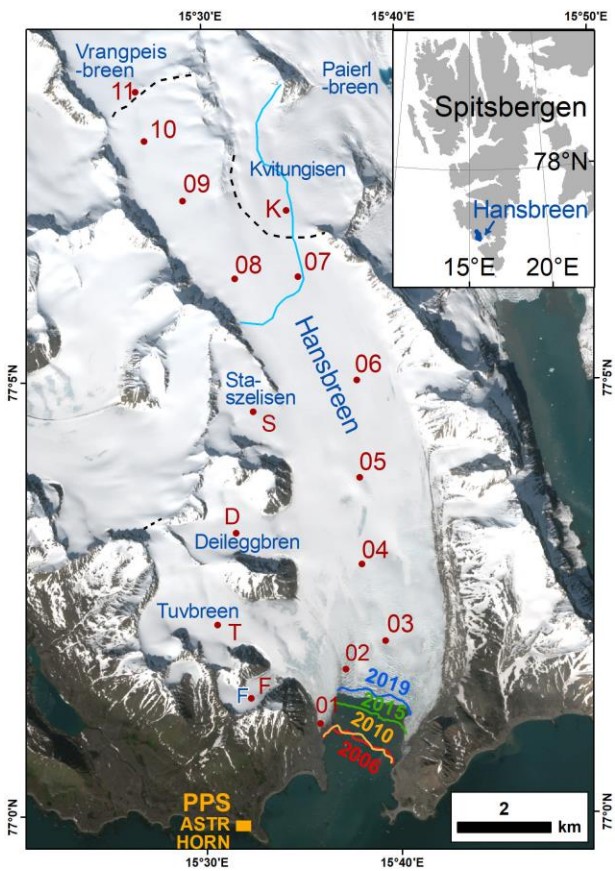

**Figure 1: Location of Hansbreen. Red dots show positions of the stakes. PPS denotes the location of the Polish Polar Station. ASTR and HORN stand for GPS reference stations, F stands for Fuglebreen. Dashed black lines indicate ice divides separating Hansbreen from other ice masses. Blue line represents the average equilibrium-line altitude of Hansbreen in 2014 (after Laska et al., 2017). Terminus extent in 2006, 2010, 2015 and 2019 is determined from Błaszczyk et al. 2020, complemented. Background: Sentinel 2 (2018-07-06).**

## 3. Instruments and methodology

### 3.1 Stake network

The datasets of ice surface velocities were estimated from repeated GNSS measurements of 16 aluminium stakes deployed on Hansbreen and its tributary and neighbouring glaciers (Fig. 1). The stakes were embedded up to 8 m deep in the glacier with a steam drill or Kovacs ice drill. The positioning of the stakes was selected so as to cover the entire elevation range of the glacier. Stakes 01–10 represent the velocity along the main trunk of Hansbreen, whereas the labelling of stakes F, T, D and S is based on the names of the tributary glaciers, i.e. Fuglebreen, Tuvbreen, Deileggbreen and Staszelisen, respectively. The positions of stakes 11 on the glacier divide and K on Kvitungisen lie beyond the current border of Hansbreen. However, we have included the data pertaining to them, as they may be crucial for better constraining numerical models of glacier dynamics or studies of ice elevation changes. The positions of the stakes were measured from summer 2006 to summer/autumn 2019.

The speed of several grounded calving glaciers tends to increase as ice thins towards the terminus (Benn and Evans, 1998). This has been observed on several Svalbard glaciers, e.g, Hansbreen (Vieli et al., 2000), Nordenskiöldbreen (van Pelt et al.,
2018), and Kronebreen (Schellenberger et al., 2015; Vallot et al., 2017). Due to difficult polar conditions, the frequency of measurements of individual stakes varied depending on the distance from the glacier front. The stakes were surveyed in three different time intervals:

– Stakes 01, 02, 03 and 05 were normally surveyed at 7–10 day intervals, although these were longer or shorter, as the measurements depended on the weather conditions. All these stakes were located in the ablation zone. Weekly intervals
allowed to highlight detailed variation in the glacier velocity. The stakes 01 and 02 were installed outside the main flow line of the glacier, due to the high crevassing of the frontal part.

– Stake 04 was measured once a day in order to acquire a fuller picture of the glacier's dynamics. Seasonal changes in velocities at this point are representative of the glacier's ablation zone (3.7 km from the front of the glacier in 2006, at an altitude of 192 m asl) and daily intervals allowed to highlight very short variations in glacier velocity, e.g. speed events.
– Stakes 06–11, F, T, D, S and K were surveyed roughly once a month, generally at 25–30 day intervals, although these could have been longer or shorter. For these stakes monthly velocity significantly exceeded the estimated errors, which allowed for seasonal insight in the glacier flow.

Hansbreen is a glacier with a negative climatic mass balance (-0.35ma$^{-1}$) for the studied period of 2006–2019 (WGMS –
https://wgms.ch/ latest access: 30 June 2023). The thinning rate estimated from differentiation of digital elevation models for period 2011-2017 was around 1ma$^{-1}$ in the upper parts the glacier, and up to 10 ma$^{-1}$ at the glacier front (Błaszczyk et al., 2019b). Because of the lowering of the glacier surface caused by negative mass balance and the advection of the stakes towards the terminus, some stakes were shifted from tens to hundreds of metres. This is especially true for stakes 02 and 03, which were deployed in a heavily crevassed frontal area. Moreover, some stakes were lost (broken by the wind or polar bears, or
buried under heavy snowfall) and had to be replaced. We present the original data sets here, so no correction was made to account for the stakes shift. Every significant change in the position of the stake is marked according to nomenclature described below. We differentiate each change in a stake's position using additional designations as follows:

– A, B, C – the consecutive positions of a stake that was significantly shifted with respect to the glacier elevation and/or its position relative to the centreline;
– 1, 2, 3 – the consecutive positions of a stake that was not significantly shifted in comparison with the size of the glacier and the centreline. This also applies to very small positional changes, of the order of decimetres, so that the velocities are determined only for the same stakes.

### 3.2 GNSS processing

Up to and including 2016, two–frequency (L1–/L2) GNSS–receivers (Leica GX1230) were used to measure the positions of the stakes in static mode. After 2016, a Leica GS14 with both GPS and GLONASS constellations was used for the measurements. During the measurements, the antenna of the GNSS instrument was placed on the top of the stake. Static GNSS surveys (15–30 minutes, but mostly 30 minutes with only a small portion being shorter) were further post–processed in Leica GeoOffice software. The observation duration was a compromise between the ambitious and unique objective of systematically monitoring glacier velocity in the Arctic and the practical constraints faced by observers during their survey fieldwork. The two reference points used in the post–processing were the ASTR (2006 to August 2009) and HORN (September 2009 to September 2019) permanent GNSS stations in the vicinity of the Polish Polar Station, Hornsund (Fig. 1; Węzka et al., 2010; Rajner, 2018). The baseline distance from the reference station to the stakes on the glacier ranged between 2.4 and 15.5 km. The coordinates in the WGS84 system were re–projected to UTM zone 33X. The ellipsoidal heights of the antenna were converted to geoidal heights using the EGM96 geoid and reduced to the glacier surface using field measurements of the stake height. When the frozen tip of the stake made it impossible to affix the antenna exactly on top of it, the antenna was placed at the front of the stake and the offset was measured and compensated for by the observer in a pre–processing procedure. In the final step, artefacts were identified manually and removed from the dataset.

One GNSS–receiver (Leica GX1230) was permanently deployed on stake 04, working in wake–up mode. This GNSS instrument was continuously battery–powered and recorded data during most of the study period with just a few gaps of several days for lack of power or poor positional accuracy after post–processing. Stake 04 was located in the main flow line of the glacier and outside the highly crevassed zone, which allowed for a safe maintenance and replacement of the battery, ensuring a continuous measurements of the stake position. The position of stake 04 was surveyed daily for 30 minutes at midnight (00:00 hrs UTC), although in 2010, 2017 and 2018, these measurements were also made at 22:00, 23:00 or 01:00 hrs. As mentioned above, daily measurements enabled observation of very short changes in glacier velocity.

### 3.3 Data and accuracy

The accuracy of the geodetically determined position of a stake is influenced by the accuracy of the static DGPS (Differential Global Positioning System) measurement (horizontal component 0.02–0.03 m; Stober and Hepperle, 2007; Bahr, 2015) and the accuracy of the survey. This last describes slight changes in the method of measuring a given stake, related to the frozen tip of the stake. We made no correction for the tilt of the stakes as, e.g. in Rückamp et al. (2010) or Machío et al. (2017); the observations were relatively frequent, so we assumed that this would be accounted for in the accuracy of the survey component. We assumed the measurement accuracy of a stake's position $e_x$ to be 0.12 m. This is in accordance with the accuracies reported in other similar post–processing procedures (e.g. Stocker–Waldhuber et al., 2019; Anderson et al., 2018). The error $e_x$ of stake 04 was assumed to be 0.03 m, as the antenna was stable between measurements.

The displacement error $\Delta x$ was estimated as:

$$\Delta x = \sqrt{(xi)^2 + (xi+1)^2} \, , \tag{1}$$

where $e_{xi}$ and $e_{xi+1}$ are the positioning errors of two stake positions.

The relative error of the measured short–term ice surface velocity was estimated after Rückamp et al. (2010) to be the sum of the relative error of the displacement and time:

$$\frac{\Delta v}{v} = \frac{\Delta x}{x} + \frac{\Delta t}{t} \, , \tag{2}$$

where $\Delta v/v$ is the relative error of the velocity $V$, $\Delta x$ is the error of the displacement, $x$ is the calculated displacement of the aluminium stake, $\Delta t$ is the precision of the time observation and $t$ is the time between stake measurements. The time accuracy is 6 hours, as the observer did not always measure the stakes in the same order. In the case of stake 04, the time is accurate to the nearest second, so $\Delta t/t$ is negligible here.

In addition to the horizontal velocity, we estimated the geoidal heights of the glacier surface at the position of each stake. The accuracy of the vertical component of static DGPS measurements is 0.03 m (Stober and Hepperle, 2007). However, there may be additional errors in the height reduction from the top of the stake, where the antenna was mounted, to the snow or ice horizon. As the ice horizon is difficult to define during the ablation season, we assessed the height accuracy to 0.1 m.

Apart from estimated horizontal velocities of the stakes between GNSS measurements, we provide annual and seasonal ice velocity records. These are based on the hydrological year, here defined as the period from 1 October to 30 September (Cogley et al., 2011), with summer identified as the period from 1 June to 30 September, and winter as that from 1 October to 31 May.

## 4. Datasets

### 4.1 Horizontal velocities

Throughout the period studied, the positions of the stakes were changed as described above. Figure 2 shows the locations of each stake and the field observations. The time series of the horizontal velocity calculated for each stake, together with the estimated accuracy of measurements, are given in Fig. 3. The measurements of the stakes started between 3 July and 2 August 2006. The continuous measurements of the stake positions ended at different times between June and October 2019. Stakes 02 and 03 were surveyed only until November 2018, as they were approaching the terminus of the glacier and the ice surface was too heavily crevassed for measurements to be conducted safely. There are some gaps in the data sets because a stake was lost or the accuracy of the GNSS measurement was poor. Therefore, some velocities were estimated for periods longer than described above (i.e. longer than daily, weekly or monthly). Although the longer observation period assures higher accuracy of measurements (see formula 2), it can cause some short speed changes to remain unnoticed in the observation.

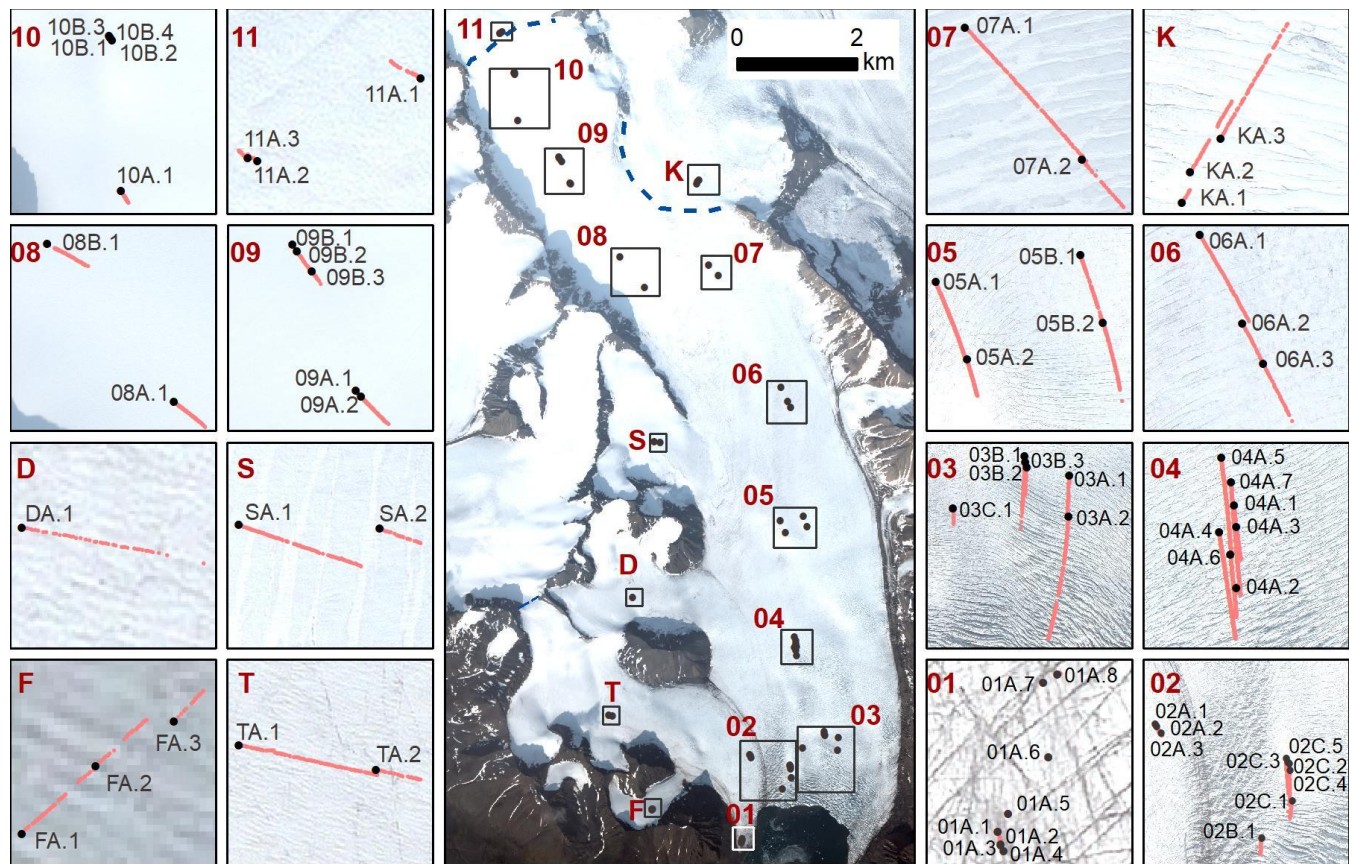

**Figure 2. Location of all surveyed stakes. Black dots represent the first position of each new dataset and light red dots show all measurements. Dashed blue navy lines indicate ice divides separating Hansbreen from other ice masses. Background: Pleiades (2017-08-20; Błaszczyk et al. 2019b).**

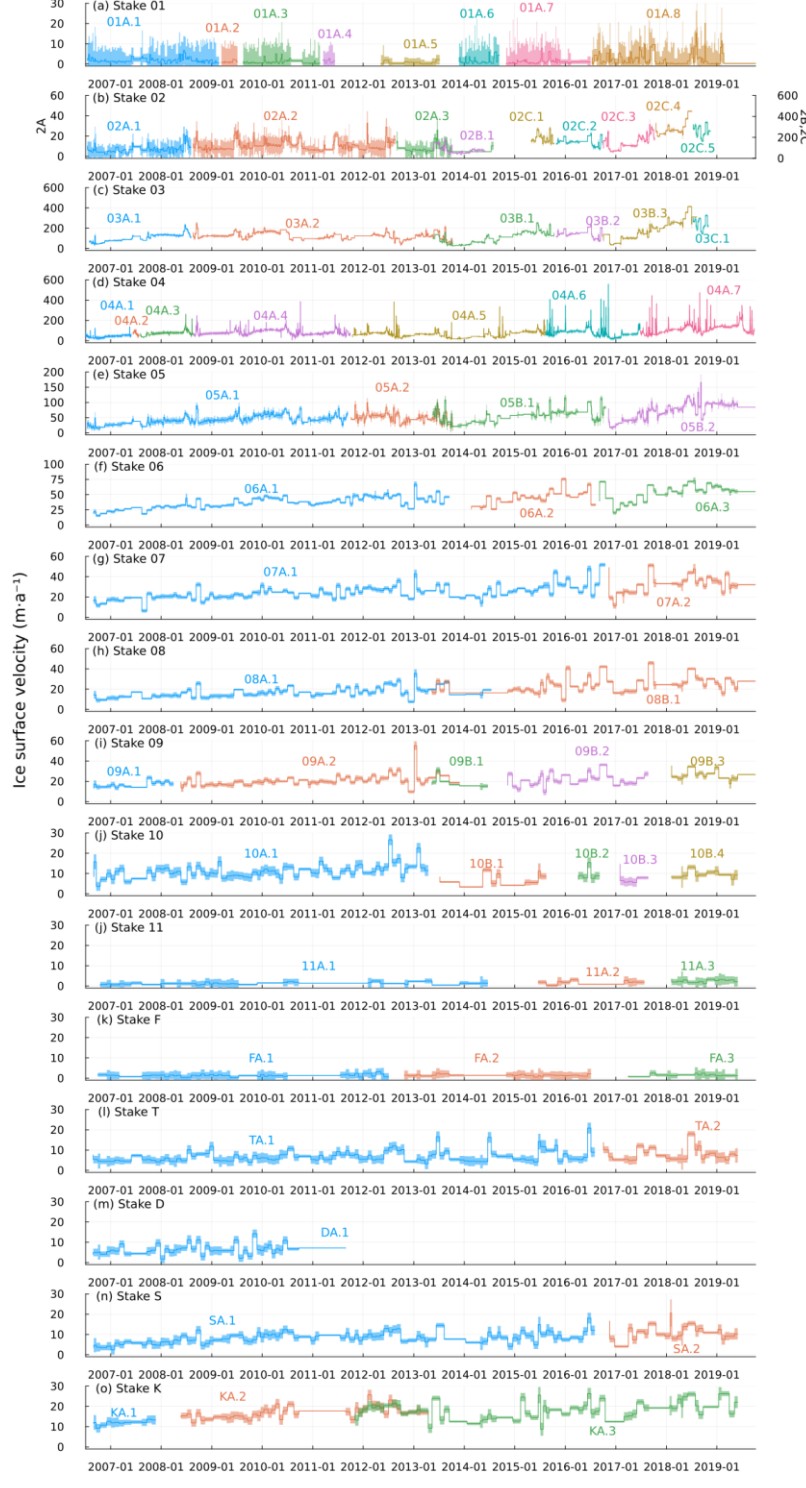

**Figure 3. Time series of horizontal velocities for all stakes from 2006 to 2019. Shaded areas indicate accuracy ranges. Note different y axis for stake 02.**

The median values of velocities estimated from all the observations on each stake are listed in Fig. 4. Stake 01 was situated near the glacier terminus, on stagnant ice, i.e. part of a glacier that does not flow at a detectable rate (Cogley et al., 2011). This stake moved very slowly towards the front until 2013 (stakes 01A.1 – 01A.5). Since the end of 2013, this stake has been almost stationary, strongly influenced by the retreating ice front. Surveys repeated every few days yielded distances between consecutive measurements of the order of one to a few centimetres, which is well below the accuracy of the measurements

(Fig. 3). Owing to the small displacements and the short times between the measurements, the relative velocity error at stake 1 could have been as high as ±2600%. Similarly, the velocities of stakes 02A (up to 2013), 11 and F are lower than the accuracy of surveys. The relative velocity error for these stakes was as high as 200%, and those stakes were mostly surveyed every few days. Therefore, we wish to stress that these data should be used with a monthly or annual temporal resolution, as the surface velocity of a glacier depends upon the interval of time between consecutive measurements (Meier, 1960).

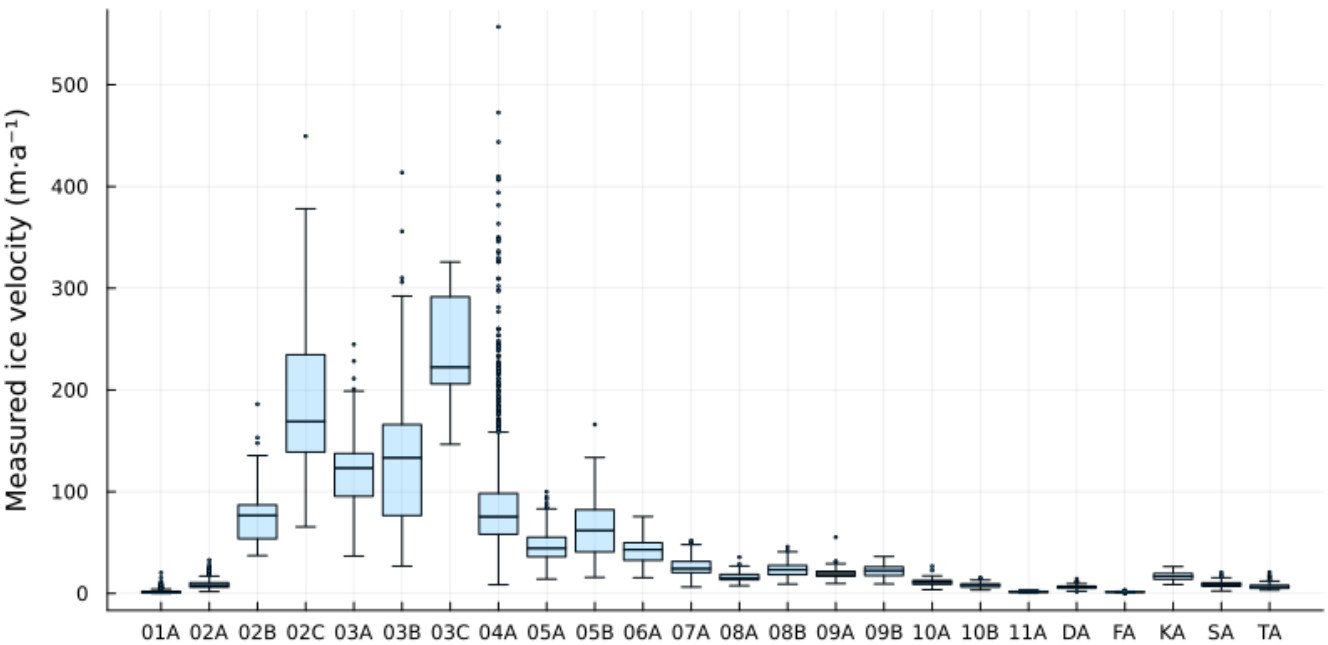

**Figure 4. A box-and-whisker plot of measured ice velocities at Hansbreen in 2006-2019: a horizontal line is the median ice velocity, box edges represent the first and third quartiles, the whiskers correspond to a minimum and maximum velocity excluding outliers, and the outliers are presented as dots.**

The highest velocity was recorded for stakes 02C, 03A, 03B and 03C (Fig. 4). The variability of the velocity was also the

220 highest at these stakes. Their positions changed significantly a few times, at different distances from the Hansbreen glacier terminus, as this retreated by c. 1 km during the study period. Hence we stress here, that especially in the case of those stakes close to the front, caution is advised when drawing long–term conclusions, which should be analysed in the light of increasing surface flow velocities towards the calving front (Pillewizer, 1939; Krimmel, 1992; Vieli et al., 2000).

The velocities of stake 04 represent a unique dataset with a very high temporal resolution (Fig. 3c). Such a time series enables one to visualise several short speed events, representing a considerable increase in glacier velocity (Kamb et al., 1994; Meier et al., 1994; Vieli et al., 2004). Such speed events were apparent not only at stake 04 measured every 24 h, but also at the other stakes measured every few days, and even at monthly intervals when the acceleration was higher than the accuracy of the measurements. However, there are some gaps in the dataset due to the poor accuracy of GNSS measurements, the lack of power or instrument failure. Continuous monitoring of stake 04 started on 3 July 2006 and ended on 29 September 2019. There are no observations for a total of 1256 days, i.e. 26% of the entire period. In the majority of cases, the data gaps are no longer than one to a few days. There are 50 data gaps of 6 days or longer, mainly in 2013 and 2014. The longest interruptions in the data measurements were 38 days in October–November 2013 and 48 days in March–April 2014. Owing to terminus retreat and glacier movement, the distance of stake 04 from the front decreased from c. 3.7 km in 2006 to c. 2.7 km in 2019. Hence, as in the case of stakes 02 and 03, long–term conclusions should be analysed in the light of increasing of surface flow velocities towards the calving front.

## 4.2 Accuracy assessment at stake 01

Stake 01 was deployed on stagnant ice (Cogley et al., 2011) that had terminated on the land and remained almost motionless since the end of 2013; hence, velocity measurements provide an excellent source for assessing the accuracy of measurements of stake positions (Fig. 8). Stake 01A.6 moved very slowly to the east, shifting 0.23 m between November 2013 and September 2014. Stake 01A.7 moved northwards, in the opposite direction to that of the glacier, shifting 0.46 m between October 2014 and July 2016. It is difficult to separate the slow movement of a stake from its tilting over a longer period or from the influence of the direct vicinity of the steep front of the glacier, and therefore from the influence of tensions due to intensive surface melting, dry calving (Cogley et al., 2011) and surface cracking. The distribution of measurements of stakes 01A.6 and 01A.7 (Fig. 8) shows that the distances between consecutive measurements, usually carried out every few days, were within the measurement error assessed in this paper. This confirms the accuracy of the positional measurements assumed in this paper to be 0.12 m. The assumed accuracy includes the accuracy of the GNSS measurements (0.02–0.03 m; Stober and Hepperle, 2007; Bahr, 2015), together with all the other uncertainties related to the antenna offset compensation and stake tilting over a short time period. The data for stake 1A.8 cannot be used for such a quality assessment, as the glacier front melted and retreated considerably from 2016 to 2019, so the distance from the stake to the front decreased from about 200 to 100 m, and the exceptionally high velocity (Fig. 3a) was due to the noise caused by surface melt, opening of crevasses and dry calving of the glacier (Cogley et al., 2011).

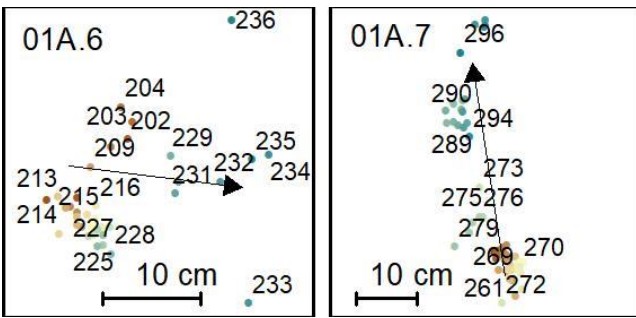

**Figure 8. Position changes of the stake 01A.6 and 01A.7. The numbers and colours (from red to dark green) indicates subsequent measurements.**

### 4.3 Annual and seasonal velocities

In order to encourage the wider use of the data presented in this paper, we also give the annual (hydrological year) and seasonal velocities derived from continuous measurements. These velocities were estimated as the weighted average of all measurements, the weights being proportional to the duration of the measurement interval. The most common problems experienced were discontinuities in the stake position surveys, due to instrument failure or logistical problems with reaching the stake (e.g. safety reasons related to crevasses, insufficient light during the polar night, or harsh weather conditions). We therefore estimated the annual and seasonal velocities only for periods with continuous datasets. The timing of a seasons was established according to calendar dates, whereas conducting field measurements was limited by the weather and snow conditions. Therefore, if the beginning/end of the year or season fell in between two field measurements, the velocity was calculated as weighted in proportion to the length of the period falling within a particular season.

Data series prepared in this way can be used to assess the spatiotemporal pattern of the Hansbreen flow. The spatial pattern of the Hansbreen velocity and the increase in velocity towards the calving front has been the subject of previous studies (Pillewizer, 1939; Jania, 1988; Vieli et al., 2004). Here, to exemplify the dataset, we show the mean annual horizontal velocities for all the stakes in the hydrological year 2006–2007 (Fig. 5). The velocity increases from the upper parts of Hansbreen (stakes 09A and 10A) towards the terminus. The highest annual velocity (in 2006/2007) was recorded at stakes 03A and 04A, i.e. those nearest to the front. The stakes deployed on the tributary glaciers revealed very slow movement, typically of the order of just a few metres per year. A similar velocity was calculated for stake 11, which flows in the opposite direction, away from the main tongue of Hansbreen, i.e. towards Vrangpeisbreen. The ice at stake K flows towards Kvitungisen into the neighbouring Paierlbreen glacier system. Although stakes 11 and K are located outside the Hansbreen basin, they can help in a broader understanding of changes in glacier dynamics and the surface height at the catchment divide, particularly the possible ice flow piracy (McCormack et al., 2023) between Hansbreen and Paierlbreen.

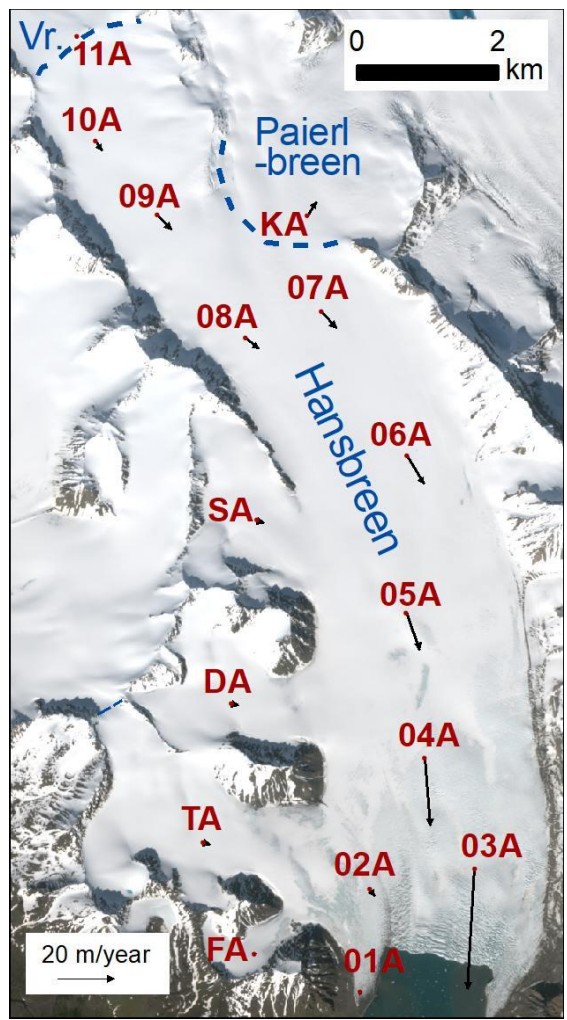

**Figure 5. Mean annual horizontal velocities for the hydrological year 2006-2007, as an example of the spatial pattern of Hansbreen velocity. The stakes 11 and K flow into other ice masses. Vr. stands for Vrangpeisbreen. Background: Sentinel 2 (2018-07-06).**

The data show a homogeneous pattern of annual and seasonal increases and decreases in velocities recorded at all the stakes, although the maximum and minimum velocities at particular stakes were noted in different years (Fig. 6). For example, the largest annual velocity was recorded in 2009 and 2017–2018 (stakes 04 – 08) and in 2011 and 2017 (stakes 10, K, SA, TA – no data for 2018). The slowest annual velocities were noted in 2006/2007 and 2013/2014 at stakes 04 and 05, but in 2006/2007– 2008/2009 and 2013/2014 at stakes farther up the glacier. The highest summer velocities were generally recorded in the 2013/2014 and 2017/2018–2018/2019. A positive trend in velocities at both the annual and seasonal scales (Fig. 6) can be inferred for stakes that did not shift significantly during the measurement period (see Fig. 2). There was a similar positive trend in velocities at stake K, which is beyond the current border of Hansbreen where the ice flows into the Paierlbreen glacier. There was a similar increase at stake K, which is beyond the border of Hansbreen, where this flows into the Paierlbreen glacier.

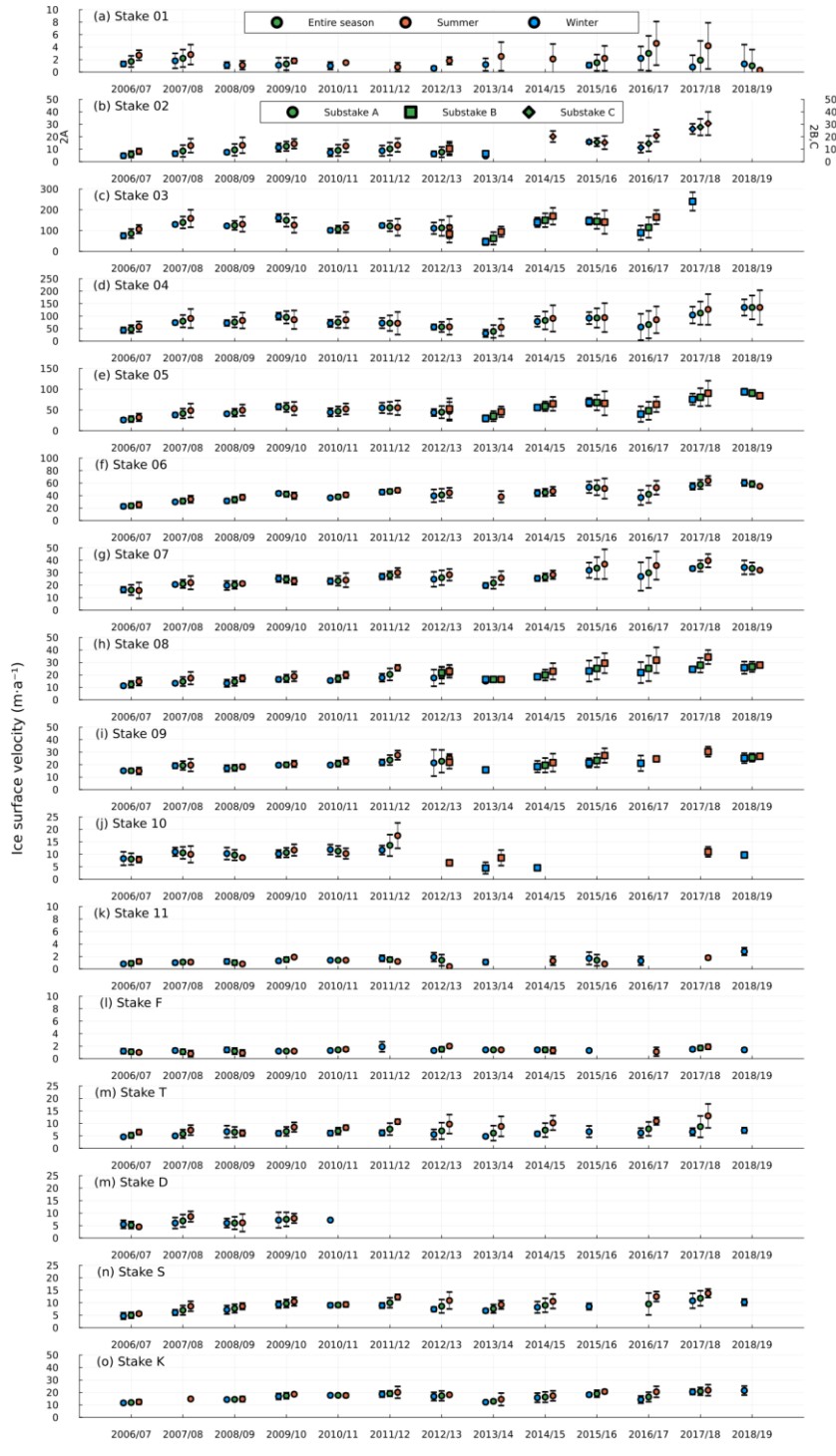

**Figure 6. Annual and seasonal velocities determined for hydrological years, together with standard deviations.**

There was significant variability in both the annual and seasonal velocities of all the stakes between the years. At particular stakes, the velocities between the fastest and slowest years differed by a factor of two, or even three in some cases. The data also show the differences between the annual, summer and winter velocities in particular years. Generally, the glacier speed exhibits a clear annual cycle, and the velocity is the highest in summer and the lowest in winter (Meier et al., 1985; Van der Veen, 1995). In some cases, however, the winter velocities are either higher (e.g. stakes 03A, 04A, 05A, 06A, 07A in 2010)

or very similar to the summer and annual velocities (e.g. stakes 08A, 09A, KA, SA in 2010). Relatively small variations between the annual and seasonal velocities are also apparent in 2014 and 2015 for the majority of stakes, and also in other years at particular stakes. Importantly, the dates of the seasons were established according to calendar dates. As glacier acceleration can also occur after the summer period, seasonal velocities should be analysed in light of such events and the meteorological and hydrological conditions (Krimmel and Vaughn, 1987; Walters and Dunlap, 1987). The largest variations

between the annual and seasonal velocities were recorded at stakes 02 and 03, i.e. those nearest to the front. But, as already mentioned, the positions of stakes shifted a few times, and fluctuations of velocities are affected by their changing distance to the glacier front (Pillewizer, 1939; Vieli et al., 2004).

## 4.4 Glacier elevation

Besides the horizontal position and velocity, the datasets include the elevation of the glacier surface. Representative datasets

of glacier surface elevation measurements at stakes 01, 05 and 10 are shown in Fig. 7, which depicts the change in surface elevation of different parts of the glacier over time. Note that these data correspond not only with emerging or submerging ice flow but also take account of changes resulting from snow accumulation and glacier ablation. This time series is not continuous, as outliers due to incorrect field measurements of the stake height were filtered from the datasets.

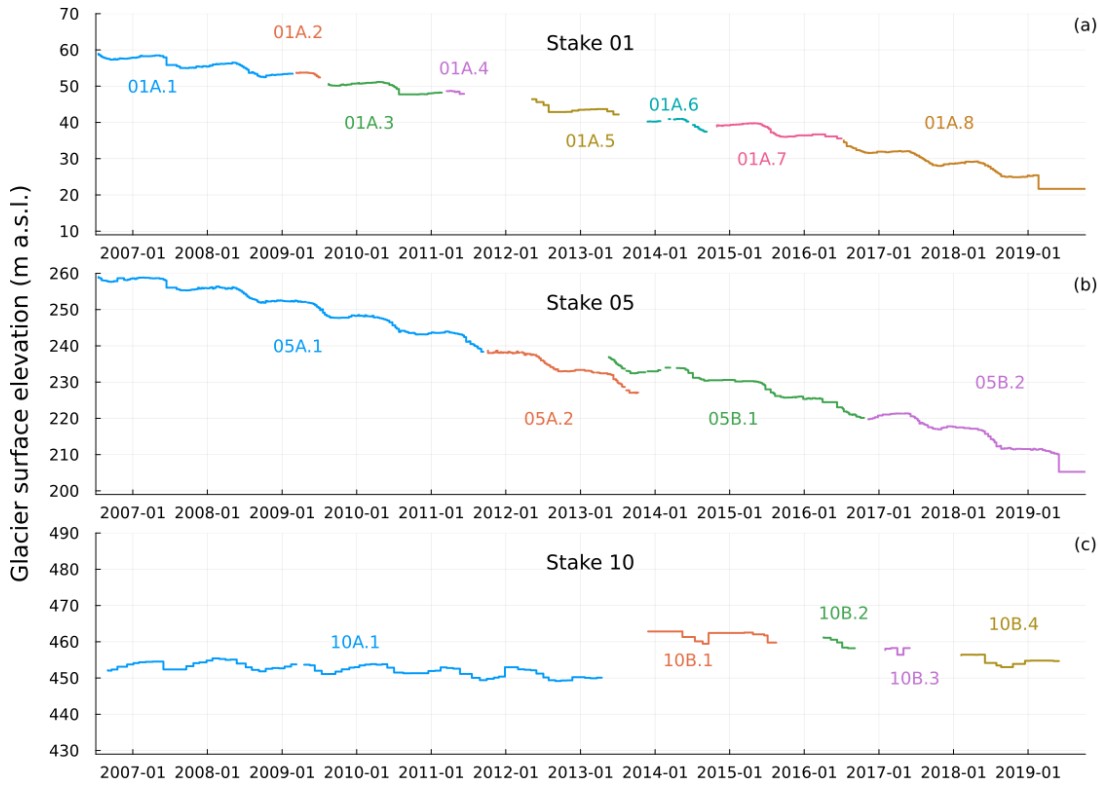

**Figure 7. Representative dataset of height measurements of stake 01, 05, 10.**

## 5. Data structure and availability

All the data have been made publicly available through the Zenodo repository: https://doi.org/10.5281/zenodo.8289380 (Błaszczyk et al., 2023). There are two .csv files for each of the 16 ablation stakes (01–11, F, T, D, S and K). The first one (e.g. *Stake_01_position.csv*) contains a time series of GNSS–derived position measurements in the UTM (zone 33X) coordinate system. This file has the following columns: sub–stake label, date of measurements, easting, northing, geoidal height (EGM96). The second one (e.g. *Stake_01_velocity.csv*) is a time series of surface ice flow velocity, and is divided into the following columns: sub–stake label, date of the first measurement, date of the second measurement, surface ice flow velocity and accuracy of the surface ice flow velocity measurement. Additionally, we provide three files with annual, summer and winter velocities estimated for a hydrological year (*Stakes_annual_velocity.csv, Stakes_summer_velocity.csv* and *Stakes_winter_velocity.csv,* respectively). These files are divided into the following columns: hydrological year, velocity for each consecutive sub–stake and standard deviation of velocity for each sub–stake. The file *Hansbreen_preprocessing_code_stakes.zip* includes the code used for velocity estimation and data preprocessing. This code was written in the Bash and Julia programming languages.

## 6. Summary

This paper presents the velocity datasets resulting from an in situ monitoring survey of the Hansbreen glacier (Svalbard) during the period 2006–2019 using the GNSS system. The dataset includes high–temporal resolution velocities at 16 stakes with accuracy assessments, as well as mean annual and seasonal velocities. This dataset is one of the longest datasets documenting glacier movement on Svalbard. The data exhibit a pattern of velocities characteristic of tidewater glaciers (Pillewizer, 1939; Krimmel, 1992; O'Neel et al., 2001), which increase from the ice divides towards the calving terminus. Further, the glacier typically accelerates during the summer months (Kamb et al., 1994; Meier et al., 1994), although higher velocities were also recorded in autumn and winter.

The shortcoming of this dataset is that it does not allow for an easy analysis of dynamic responses to climate changes, as the velocities were measured at stakes that change their position with time and represent a Lagrangian velocity field (Machío et al., 2017). Furthermore, the glacier terminus retreated about one kilometre during the study period. Thus, to investigate the relation between intraseasonal or interannual velocities and climate forcing, stake velocity should be corrected to the same position in space, and to the same distance from the glacier termini. Nevertheless, after such correction, these data might be further investigated with regard to changes in the glacier dynamics in light of climate change in the region (Wawrzyniak and Osuch, 2020), together with changes in the terminus position at different timescales (Ultee et al., 2020). Overall, these in situ ice flow velocity datasets can be used for tuning numerical models of glacier dynamics, e.g. modelling of flow of a grounded calving glacier (Vieli et al., 2000), developing glacier–plume and glacier–fjord circulation coupled models to simulate glacier-ocean system (De Andres et al., 2021), numerical simulation of the ice flow velocity in the future (e.g. Ai et al., 2019a), and changes of strain rate between neighbouring stakes (Ai et al., 2019b). Further, the data set is an invaluable source of information on short–term processes such as speed events and seasonal velocity changes (van Pelt et al., 2018). Moreover, such a data set can be of great importance for validating satellite–derived ice flow fields with in situ velocity measurements (Schellenberger et al., 2015; Fahnestock et al., 2016; Friedl et al., 2021) and digital elevation models derived from different remote sensing techniques, as previously used by Błaszczyk et al. (2019b).

## Author contributions

JJ, DP and PG conceived the project. DP, MP, MB, BL, DI, and ML maintained the stake network and the instrumentation. Data curation were provided by MB, BL, MP. Conceptualization of the paper were performed by MB, MP, JJ and BL. MB, DP and MP led data processing and analysis. MB and MP wrote the original draft, and co–authors contributed text and edits. Funding acquisition was performed by PG, BL, MP, and JJ.

**Competing Interests.** The authors declare that they have no conflict of interest.

**Acknowledgements**

The data were collected within the framework of the glaciological monitoring of the Polish Polar Station Hornsund managed by the Institute of the Geophysics Polish Academy of Sciences, subsidized by the Polish Ministry of Education and Science (Decision No. 3/524698/SPUB/SP/2022). Special thanks are due to the staff of the Institute of Geophysics PAS, the Polish Polar Station, Hornsund, and the participants of the University of Silesia's expeditions to Spitsbergen for maintaining the glaciological monitoring and conducting the fieldwork. The Pléiades used in the figures were provided by the Pléiades Glacier

Observatory (PGO) programme of the French Space Agency (CNES), and Sentinel–2 was obtained from Copernicus Sentinel data (2018). The research and logistical equipment of the Polar Laboratory of the University of Silesia in Katowice was used during the fieldwork. The publication was created as part of a project co-financed by the Minister of Education and Science under contract No. 2023/WK/02. BL was supported by a subsidy from the Polish Ministry of Education and Science for the Institute of Geophysics, Polish Academy of Sciences. MP was supported by a grant from the Priority Research Area

(Anthropocene) under the Strategic Programme Excellence Initiative at the Jagiellonian University. Publication co-financed by funds allocated under the Research Excellence Initiative of the University of Silesia in Katowice.

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
