# Peer review of "High temporal resolution records of the velocity of Hansbreen, a tidewater glacier in Svalbard"

_Earth System Science Data, 2023_

## Referee Comment (RC1)

The manuscript titled **'High Temporal Resolution Records of the Velocity of Hansbreen, a Tidewater Glacier in Svalbard'** presents a significant record and of glacier surface ice flow velocities, collected in situ using GNSS technology from 2006 to 2019. The study focuses on the Hansbreen glacier in Svalbard and provides a unique viewpoint on the movement patterns of this tidewater glacier. Here's an overall assessment:

The manuscript contributes to the field of glaciology by providing a wide-ranging dataset of ice-flow velocities. It addresses the significance of understanding glacier dynamics in the context of climate change and the need for in situ measurements. The manuscript is well-structured and comprises relevant background information, a detailed study area, an explanation of the methodology, and clear findings. The manuscript presents a well-documented dataset with a high temporal resolution, which is one of its primary strengths. The use of GNSS technology and the inclusion of accuracy assessments contribute to the reliability of the data. The study area, Hansbreen, is a well-studied tidewater glacier in Svalbard, making it relevant for broadening the understanding of glacier dynamics and their response to climate change. The preprint provides the stake network, GNSS processing, and how annual and seasonal velocities were calculated, ensuring that readers can comprehend the methodology. Relevant prior studies are cited throughout the manuscript, enhancing the paper's credibility and grounding it in the existing body of glacier research.

**Recommendation to the Editor:**

Based on the review, it is recommended to consider the manuscript for acceptance with minor revisions. The study provides a sound methodology and meaningful perceptions into glacier dynamics. The minor revisions primarily relate to improving the manuscript's organization, discussing data analysis implications, and emphasizing the relevance of the findings within the broader field of glaciology and climate science.

The only issue is that 15~30 minutes observations using GNSS can not always get fixed solutions for the baseline processing, which will lead to lower spatial accuracy. In order to get more robust positioning results, a little bit longer observation period (such as 40 minutes) is recommended for future field GNSS measurements, especially for the stakes far away from the base GNSS station.

As to the dataset publicly available through the Zenodo repository, only processed stake locations are presented. If the original or RINEX data of GNSS were available, we can re-calculate the stake positions with other software besides Leica GeoOffice, then more robust solutions are possible.

**Improvement Suggestions in the manuscript**

❖ While the introduction highlights the significance of long-term in situ measurements, it lacks a clear statement of the specific research objectives or questions the study aims to address. How about the sampling rate in field GNSS measurements?

❖ Consider providing more context about why Hansbreen in Svalbard is a suitable study area and how it fits into the broader field of glaciology. Explicitly state the research gap that this study intends to fill.

❖ The transition from general background information to the focus on Hansbreen could be made smoother.

❖ The final sentence about 'our understanding of the physics of glacier motion' is to some extent unclear. Clarify what specific visions this study may contribute to this understanding.

❖ Provide more context about the significance of these geographical and physical characteristics in section 2. Study area. How do these details relate to the objectives of the study or the broader field of glaciology?

❖ Include geographic coordinates in study area section and to locate Hansbreen on a map.

❖ It would be helpful to provide more context about why specific stakes were surveyed at different time intervals and how these intervals were determined. What impact do these intervals have on data accuracy and interpretation?

❖ Mention how the placing of stakes that shifted significantly was addressed in data analysis. Were corrections made to account for these shifts?

❖ When referencing previous studies (Błaszczyk et al., 2019b; W GMS), include more specific details about the findings or relevance of these studies in the context of this manuscript methodology.

- ❖ The continuous operation of the GNSS instrument on stake 4 is noteworthy, but it would be beneficial to explain why this particular stake required continuous monitoring and the specific insights gained from it.
- ❖ The information about measurement times (measurements at midnight) is mentioned but could be linked to the objectives or data analysis.
- ❖ When discussing the challenges related to discontinuities in stake position surveys, consider briefly mentioning how these issues were addressed or mitigated to maintain data accuracy.
- ❖ Given that stakes 11 and K have unique flow directions, consider explaining the relevance of this information in the broader understanding of glacier dynamics in the region.
- ❖ When discussing the challenges and limitations of the dataset, consider elaborating on how these challenges might impact the dataset's use and interpretation.
- ❖ Highlight the practical implications of the dataset, such as its applications in tuning numerical models, validating satellite-derived ice flow fields, and assessing short-term and seasonal glacial processes, more clearly.

---

## Author Response (AR1)

Dear Editors of the of the Earth System Science Data,

We are pleased to resubmit a manuscript entitled "High temporal resolution records of the velocity of Hansbreen, a tidewater glacier in Svalbard".

We appreciate the comments provided by the reviewers, as they provided valuable feedback to enhance the impact of our work. In reference to our responses dated December 4, 2023, we have integrated all comments from RC1 and one comment from RC2 into the manuscript. The manuscript has been revised in accordance with the given suggestions. We have edited the text, corrected a figure, and included additional descriptions based on the comments. Detailed answers are included in responses to Reviewers.

We remain in contact in case of additional questions or doubts.

Sincerely,
On behalf of the authors,
Małgorzata Błaszczyk

**Reviewer 1**

The manuscript titled **'High Temporal Resolution Records of the Velocity of Hansbreen, a Tidewater Glacier in Svalbard'** presents a significant record and of glacier surface ice flow velocities, collected in situ using GNSS technology from 2006 to 2019. The study focuses on the Hansbreen glacier in Svalbard and provides a unique viewpoint on the movement patterns of this tidewater glacier. Here's an overall assessment:
The manuscript contributes to the field of glaciology by providing a wide-ranging dataset of iceflow velocities. It addresses the significance of understanding glacier dynamics in the context of climate change and the need for in situ measurements. The manuscript is well-structured and comprises relevant background information, a detailed study area, an explanation of the methodology, and clear findings. The manuscript presents a well-documented dataset with a high temporal resolution, which is one of its primary strengths. The use of GNSS technology and the inclusion of accuracy assessments contribute to the reliability of the data. The study area, Hansbreen, is a well-studied tidewater glacier in Svalbard, making it relevant for broadening the understanding of glacier dynamics and their response to climate change. The preprint provides the stake network, GNSS processing, and how annual and seasonal velocities were calculated, ensuring that readers can comprehend the methodology. Relevant prior studies are cited throughout the manuscript, enhancing the paper's credibility and grounding it in the existing body of glacier research.

**Recommendation to the Editor:**

Based on the review, it is recommended to consider the manuscript for acceptance with minor revisions. The study provides a sound methodology and meaningful perceptions into glacier dynamics. The minor revisions primarily relate to improving the manuscript's organization, discussing data analysis implications, and emphasizing the relevance of the findings within the broader field of glaciology and climate science.

The only issue is that 15~30 minutes observations using GNSS can not always get fixed solutions for the baseline processing, which will lead to lower spatial accuracy. In order to get more robust positioning results, a little bit longer observation period (such as 40 minutes) is recommended for future field GNSS measurements, especially for the stakes far away from the base GNSS station.

As to the dataset publicly available through the Zenodo repository, only processed stake locations

are presented. If the original or RINEX data of GNSS were available, we can re-calculate the stake positions with other software besides Leica GeoOffice, then more robust solutions are possible.

More detailed information, please see the uploaded pdf document.

Dear Prof. Songtao Ai,

We sincerely appreciate your valuable comments and suggestions. In reference to our response dated December 4, 2023, we have incorporated all of the comments into the manuscript. We have edited the text, corrected a figure, and included additional descriptions based on the provided suggestions. The number of lines refers to the document with tracked changes.

**Improvement Suggestions in the manuscript**

While the introduction highlights the significance of long-term in situ measurements, it lacks a clear statement of the specific research objectives or questions the study aims to address. How about the sampling rate in field GNSS measurements?
We have highlighted the study objectives in lines 50-56.

Consider providing more context about why Hansbreen in Svalbard is a suitable study area and how it fits into the broader field of glaciology. Explicitly state the research gap that this study intends to fill.
Please see the answer below.

The transition from general background information to the focus on Hansbreen could be made smoother.
In response to both comments above, background information on Hansbreen has been relocated to next paragraph, and details regarding the representativeness of Hansbreen and the research gap this study intends to fill have been incorporated in lines 50-63.

The final sentence about 'our understanding of the physics of glacier motion' is to some extent unclear. Clarify what specific visions this study may contribute to this understanding.
It has been clarified in lines 70-73.

Provide more context about the significance of these geographical and physical characteristics in section 2. Study area. How do these details relate to the objectives of the study or the broader field of glaciology?
An additional description of the geographical and physical characteristics of the study area has been added in lines 86-95.

Include geographic coordinates in study area section and to locate Hansbreen on a map.
Geographical coordinates have been added to the text (line 76) and Fig. 1.

It would be helpful to provide more context about why specific stakes were surveyed at different time intervals and how these intervals were determined. What impact do these intervals have on data accuracy and interpretation?
Suggested information has been added in lines 114-128.

Mention how the placing of stakes that shifted significantly was addressed in data analysis. Were corrections made to account for these shifts?
It has been clarified in lines 137-138.

When referencing previous studies (Błaszczyk et al., 2019b; W GMS), include more specific details about the findings or relevance of these studies in the context of this manuscript methodology.
Suggested information has been added in lines 130-133.

The continuous operation of the GNSS instrument on stake 4 is noteworthy, but it would be beneficial to explain why this particular stake required continuous monitoring and the specific insights gained from it.
Please see the answer below.

The information about measurement times (measurements at midnight) is mentioned but could be linked to the objectives or data analysis.
In response to the both comments above, information clarifying the significance of measurements on stake 04 has been added in lines 163-167.

When discussing the challenges related to discontinuities in stake position surveys, consider briefly mentioning how these issues were addressed or mitigated to maintain data accuracy.
The paragraph was reorganized slightly, and additional text has been included to address the above suggestions (lines 202-208).

Given that stakes 11 and K have unique flow directions, consider explaining the relevance of this information in the broader understanding of glacier dynamics in the region.
It has been explained in lines 288-290.

When discussing the challenges and limitations of the dataset, consider elaborating on how these challenges might impact the dataset's use and interpretation.
Suggested explanation has been added in lines 353-355.

Highlight the practical implications of the dataset, such as its applications in tuning numerical models, validating satellite-derived ice flow fields, and assessing short-term and seasonal glacial processes, more clearly.
The text has been modified in accordance with the suggestions provided (lines 358-365).

In reference to our response dated December 4, 2023, we have also underlined information on duration of GPS measurements in lines 149-151.

We express our sincere hope that the applied modification will result in the final acceptance of the revised manuscript.

Sincerely,
On behalf of the authors,
Małgorzata Błaszczyk

**Reviewer 2**

The manuscript presents high temporal resolution dataset of velocity of Hansbreen glacier, Svalbard. The velocity data was derived from continuous GPS measurements of dense network of stakes, mainly along the flowline of the glacier but also in the tributaries. The manuscript basically presents three datasets; horizontal velocities, annual and seasonal velocities and glacier elevation change.

Although the provided data is essential for understanding glacier dynamics, it should be complemented with other continuously measured data, such as mass balance, Automatic Weather

Station data, and GPR survey results, etc., along with bed elevation and ice thickness. I believe the paper's scope is too narrow in its present form.

One significant drawback is the lack of discussion on specific spatial and temporal patterns observed in the data. Incorporating additional glaciological data, like mass balance and ELA positions, would enhance the analysis against the observations.

In its current format, the paper solely presents GPS observations without a thorough statistical and geophysical analysis. While these data are valuable for ice flow modeling, their utility is limited without supplementary information on mass balance and meteorological data. I strongly suggest that the authors augment the existing velocity data with more geophysical data, conduct statistical analyses, and provide detailed discussions on specific observed patterns. Incorporating those changes requires substantial restructuring of the manuscript. Therefore, I cannot recommend the paper for publication in its current format.

Here are some specific suggestions on how certain sections can be improved with additional information:

1. Majority of Texts to Supplementary Material: Consider moving a significant portion of the text to supplementary material. The main paper should focus on data analysis and result discussions.

2. Horizontal Velocities: Explore why stake 2B to 4A experienced higher velocities. Investigate the ice and bed topography in those stake locations and assess any potential influence of topography.

3. Annual and Seasonal Velocities: Discuss why stakes at tributary glaciers exhibited slower velocities. Compare the ice thickness and slopes at these locations to other stakes. Address changes in velocities between summer and winter, considering the possible role of subglacial hydrology and the basal thermal state of the glacier. Explain why the highest summer velocities were recorded in 2013/14 and 2017/18-2018/19 and provide information on the mass balance of those years.

4. Glacier Elevation: Illuminate the emergence and submergence velocity of the glacier from these measurements.

5. Summary: Question the presence of the last paragraph in the summary. It could be more appropriately placed under a specific section or included as an appendix.

Dear Reviewer,

We sincerely appreciate your valuable comments and suggestions. In reference to our response dated December 4, 2023, we have incorporated comments nb 5 to the manuscript. We have moved the whole paragraph to chapter 4 as a new sub-chapter "4.2 Accuracy assessment at stake 01", lines 250-268. The number of lines refers to the document with tracked changes.

We express our sincere hope that the applied modification will result in the final acceptance of the revised manuscript.

Sincerely,
On behalf of the authors,
Małgorzata Błaszczyk

---

## Author Response (AR2)

Dear Editors of the of the Earth System Science Data,

We appreciate the comments provided by the editor. We have integrated all comments into the manuscript and have edited the text to made it clear.

Detailed responses to comments:

In an answer to the comments for line 108 (This is a bit unclear - longer or shorter, how much (max min)?) and lines 115-116 (same here "these intervals" and specify max and min gap).
We decided not to give the exact number of days. Each stake had it's own history of measurements and post-processing, and the number of datagaps was very different. Instead we have added more explanation regarding data-gaps at lines 203-205 (document with tracked-changes).

Figure 8: Did you include the errors in this figure?
We did not included the errors. Taking into account vertical scale, the error band of height (0.1 m) would not be visible.

Thank you for your effort. We remain in contact in case of additional questions or doubts.

Sincerely,
On behalf of the authors,
Małgorzata Błaszczyk